# Phases of the Bose–Einstein Condensate Dark Matter Model with Both Two- and Three-Particle Interactions

**Alexandre M. Gavrilik *** [ID] **and Andriy V. Nazarenko** [ID]

Bogolyubov Institute for Theoretical Physics of NAS of Ukraine, 14b, Metrolohichna Street, 03143 Kyiv, Ukraine; nazarenko@bitp.kiev.ua
* Correspondence: omgavr@bitp.kiev.ua

**Abstract:** In this paper, we further elaborate on the Bose–Einstein condensate (BEC) dark matter model extended in our previous work [*Phys. Rev. D* **2020**, *102*, 083510] by the inclusion of sixth-order (or three-particle) repulsive self-interaction term. Herein, our goal is to complete the picture through adding to the model the fourth-order repulsive self-interaction. The results of our analysis confirm the following: while in the previous work the two-phase structure and the possibility of first-order phase transition was established, here we demonstrate that with the two self-interactions involved, the nontrivial phase structure of the enriched model remains intact. For this to hold, we study the conditions which the parameters of the model, including the interaction parameters, should satisfy. As a by-product and in order to provide some illustration, we obtain the rotation curves and the (bipartite) entanglement entropy for the case of a particular dwarf galaxy.

**Keywords:** dark matter; halo; Bose–Einstein condensate; two- and three-particle self-interactions; two-phase structure; first-order phase transition; dwarf galaxies; rotation curves; entanglement entropy

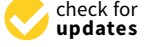



## 1. Introduction

Although the concept of dark matter (DM) is a widely accepted one, its precise nature still needs elucidating. There exists a vast multitude of different approaches and models, among which the modeling of DM as Bose–Einstein condensate (BEC), as can be seen, e.g., in [1–4], the overviews [5,6] and many others, finds each time more and more support. Positions of the BEC model of DM were especially enforced after the works [7–9] which demonstrated the ability of BEC DM to avoid the core-cusp and gravitational collapse [10] problems. Furthermore, the model gives a quite successful description [7,11–13] of the rotation curves of a number of galaxies—at least the dwarf and low surface brightness ones.

Nevertheless, even within this well-elaborated model, there also exist some tensions and issues which can be improved. To this end, there is in particular a possibility to apply appropriate tools from the powerful and efficient theory of deformations. Namely, the $\mu$-deformed analog of the Bose-gas model developed in [14], with the so-called $\mu$-calculus as a base, has clearly manifested the following preferable features: (i) the evaluated mass of a DM halo appears more realistic; (ii) the obtained critical temperature of the condensation of $\mu$-Bose gas $T_C^{(\mu)}$ depends on the deformation parameter $\mu$, $\mu > 0$, and is higher [15] than the usual $T_C$; (iii) the $\mu$-deformation-based description of the rotation curves [16] fits better than the curves inferred within the ordinary BEC model.

It is also worth mentioning the recent work [17] which uses the concept of deformed spatial commutation relations for a scalar field in order to develop a class of generalization of the Bose-condensate DM model. Such an extension has good potential to achieve improvements.

Another line of extension of the BEC model of DM, developed recently in [18], involves the sixth-order (or three-particle) self-interaction term $\psi^6$. Due to presence of the latter,

the modified model discloses a nontrivial phase structure: there exist two distinct phases, a certain region of instability, and the possibility of first-order phase transition.

In order to make the extended model even more complete, it is natural to include, besides the $\psi^6$, also the two-particle self-interaction encoded in the term $\psi^4$. The analysis of such a "doubly-nonlinear" extension of [18] and the BEC model of DM is the goal of the present paper.

The structure of the paper is the following. The necessary details of the model are given in Section 2, and the main part that involves obtaining thermodynamic functions and their key properties is presented in Section 3. In Sections 4 and 5, we briefly consider, again for the situation of the presence of both two- and three-particle interactions, the rotation curves of a selected galaxy and the respective bipartite entanglement entropy of two centrally symmetric regions of the halo of this same galaxy. In our concluding section, we present a discussion of the results.

## 2. The Model

We describe the Bose–Einstein condensate by a real function $\psi(r)$ of the radial variable $r = |\mathbf{r}|$, using a constant chemical potential $\tilde{\mu}$. Our study was based on the energy functional $\Gamma$ in a ball $B = \{\mathbf{r} \in \mathbb{R}^3 \,|\, |\mathbf{r}| \leq R\}$ and the Poisson equation:

$$\Gamma = 4\pi \int_0^R \left[ \frac{\hbar^2}{2m} (\partial_r \psi(r))^2 + m\psi^2(r) V_{\mathrm{gr}}(r) + \frac{U_2}{2} \psi^4(r) + \frac{U_3}{3} \psi^6(r) - \tilde{\mu}\psi^2(r) \right] r^2 \,\mathrm{d}r, \quad (1)$$

$$\Delta_r V_{\mathrm{gr}}(r) = 4\pi G m |\psi(r)|^2, \quad (2)$$

where $\Delta_r$ is the radial part of the Laplace operator that acts as

$$\Delta_r f(r) = \partial_r^2 f(r) + \frac{2}{r} \partial_r f(r), \quad (3)$$

$$\Delta_r^{-1} f(r) = -\frac{1}{r} \int_0^r f(s) s^2 \,\mathrm{d}s - \int_r^R f(s) s \,\mathrm{d}s, \quad (4)$$

$R$ being the radius of the ball where the matter is located.

Focusing here on the effects of interparticle interactions, we leave aside the slow rotation of the condensate [13] which can be taken into account in the chemical potential [17].

For convenience, let us introduce dimensionless variables:

$$\psi(r) = \sqrt{\varrho_0}\, \chi(\xi), \quad r = r_0\, \xi, \quad u = \tilde{\mu}\, \frac{mr_0^2}{\hbar^2},$$

$$A = 4\pi \frac{Gm^3 \varrho_0 r_0^4}{\hbar^2}, \quad Q = U_2 \frac{\varrho_0 m r_0^2}{\hbar^2}, \quad B = U_3 \frac{\varrho_0^2 m r_0^2}{\hbar^2}. \quad (5)$$

Here, $\chi(\xi)$ is a real dimensionless scalar field; $\varrho_0$ and $r_0$ characterize the *typical measures* of the central particle density and system size, respectively.

Thus, we arrive at:

$$\frac{\Gamma}{\Gamma_0} = \int_0^{\xi_B} \left[ \frac{1}{2} (\partial_\xi \chi)^2 - u\chi^2 + A\chi^2 \varphi + \frac{Q}{2} \chi^4 + \frac{B}{3} \chi^6 \right] \xi^2 \,\mathrm{d}\xi, \quad \Gamma_0 = \frac{4\pi \hbar^2 r_0 \varrho_0}{m},$$

$$\Delta_\xi \varphi(\xi) = \chi^2(\xi), \quad (6)$$

where $\xi_B = R/r_0$, whereas $\Delta_\xi$ and $\Delta_\xi^{-1}$ are given by (3) and (4) in terms of $\xi$ replacing $r$.

To estimate the range of parameter values, we turn to astrophysical situations. Since we suggest taking into account the three-particle interaction in the relatively dense DM of light bosons with masses of the order of $10^{-22}$ eV $c^{-2}$, ranges of the parameters can be found by considering the DM of galactic cores with a central mass density $\rho_0 = m\varrho_0$ of the

order of $10^{-20}\,\mathrm{kg\,m^{-3}}$, in the region of radius $r_0$ smaller than 1 kpc. Then, extracting $r_0$ from the definition (5) of the measure of the gravitational interaction $A$:

$$r_0 \simeq 0.824\,\mathrm{kpc}\left[\frac{A}{10}\right]^{1/4}\left[\frac{mc^2}{10^{-22}\,\mathrm{eV}}\right]^{-1/2}\left[\frac{\rho_0}{10^{-20}\,\mathrm{kg\,m^{-3}}}\right]^{-1/4}, \tag{7}$$

we can adopt that $A \sim 10$ [18].

It is clear that the gravity results from the integral effect of a whole system. However, the (thermo)dynamics of internal processes is determined by repulsive interactions among bosons, represented by the parameters $Q$ and $B$ under the condition $B > A$. The role of pairwise interaction, controlled by $Q$, is assumed to be comparable with the effect of gravity.

Using (7), the characteristic energy density $\varepsilon_0 = \hbar^2\varrho_0/(mr_0^2)$ is evaluated as

$$\varepsilon_0 \simeq 33.82\,\mathrm{eV\,cm^{-3}}\left[\frac{A}{10}\right]^{-1/2}\left[\frac{mc^2}{10^{-22}\,\mathrm{eV}}\right]^{-1}\left[\frac{\rho_0}{10^{-20}\,\mathrm{kg\,m^{-3}}}\right]^{3/2}. \tag{8}$$

In the pressure units, $33.82\,\mathrm{eV\,cm^{-3}} \simeq 5.42 \times 10^{-12}\,\mathrm{Pa}$.

The extremizing of the functional $\Gamma$, i.e., $\delta\Gamma/\delta\chi(\xi) = 0$, yields the set of field equations:

$$\frac{1}{2}\Delta_\xi\chi + u\chi - A\chi\varphi - Q\chi^3 - B\chi^5 = 0, \qquad \Delta_\xi\varphi = \chi^2. \tag{9}$$

We combine the model equations in the spirit of [18] by introducing the field $v(\xi)$:

$$v(\xi) = \int_0^\xi \chi^2(s)\,s\,\mathrm{d}s, \qquad v(\xi_B) = -\varphi(0). \tag{10}$$

As a result:

$$2\frac{\Gamma}{\Gamma_0} = \int_0^{\xi_B}\left[(\partial_\xi\chi)^2 - u_*\chi^2(\xi) + \frac{Q_*}{2}\chi^4(\xi) + \frac{B_*}{3}\chi^6(\xi)\right]\xi^2\,\mathrm{d}\xi$$

$$- \frac{A_*}{2}\int_0^{\xi_B}[v(\xi_B) - v(\xi)]^2\,\mathrm{d}\xi, \tag{11}$$

$$\Delta_\xi\chi + v\chi - \chi\frac{A_*}{\xi}\int_0^\xi v(s)\,\mathrm{d}s - Q_*\chi^3 - B_*\chi^5 = 0, \tag{12}$$

$$\partial_\xi v(\xi) = \xi\,\chi^2(\xi), \qquad v(0) = 0, \tag{13}$$

$$v = A_*v(\xi_B) + u_*, \tag{14}$$

where $A_* = 2A$, $Q_* = 2Q$, $B_* = 2B$, and $v$ (put instead of $u_* = 2u$) are arbitrary positive parameters. The system boundary $\xi_B$ is defined from the condition $\chi(\xi_B) = 0$ and is the *first zero* of the oscillating function $\chi(\xi)$.

In order to find a decreasing solution $\chi(\xi)$ for the admissible $\xi$ with a finite initial value $\chi_0 = \chi(0) < \infty$, we first impose $\chi'(0) = 0$ and then formulate the conditions which allow to fix $\chi_0$ through expanding $\chi(\xi) = \chi_0 + C_2\xi^2 + \dots$ at $\xi \to 0$. On substituting that in (12) and (13), the following algebraic equations result:

$$6C_2 + v\chi_0 - Q_*\chi_0^3 - B_*\chi_0^5 = 0,$$

$$vC_2 - \frac{A_*}{6}\chi_0^3 - 3Q_*\chi_0^2C_2 - 5B_*\chi_0^4C_2 = 0. \tag{15}$$

Combining the two, we find that the initial value $\chi_0$ should satisfy the equation $S(A_*, B_*, Q_*, v, \chi_0) = 0$, where:

$$S(A_*, B_*, Q_*, v, z) = A_*z^2 - (5B_*z^4 + 3Q_*z^2 - v)(v - Q_*z^2 - B_*z^4), \tag{16}$$

plus the condition $2C_2 = \chi''(0) \leq 0$. Taken altogether, these constrain $\chi_0$ as $z_1 < \chi_0 < z_2$, where:

$$z_1 = \left[ \sqrt{\left(\frac{3Q_*}{10B_*}\right)^2 + \frac{\nu}{5B_*}} - \frac{3Q_*}{10B_*} \right]^{1/2}, \quad z_2 = \left[ \sqrt{\left(\frac{Q_*}{2B_*}\right)^2 + \frac{\nu}{B_*}} - \frac{Q_*}{2B_*} \right]^{1/2}. \quad (17)$$

Technically, the search for the initial value $\chi_0$ of the model which takes into account the pair interaction is similar to the problem with three-particle interaction only [18]. Likewise, we notice three regimes (for a given $A_*$, $Q_*$, $B_*$ and $\nu$): (1) no solution for $\chi_0$ that, in Equation (12), leads to $\chi(\xi) = 0$; (2) a single solution $\chi_0$ that corresponds to a *minimal admissible value* $\nu_{\min}$ from which the system starts to evolve; (3) a pair of (positive) solutions for $\chi_0$, when we should choose a minimal one, because the other leads to divergent $\chi(\xi)$. Usually, for fixed $(A_*, Q_*, B_*)$, but increasing $\nu$, the indicated sequence of all three options occurs.

It is useful to analyze the system from the quantum-mechanical point of view. Equation (12) for $\xi \leq \xi_B$ can be conveniently rewritten in the Schrödinger form to describe the scattering of a particle whose wave function[1] is taken as $f(\xi) = c\chi(\xi)$ ($c$ is a normalization depending on a total number of particles) in the potential $V_{\text{eff}}$:

$$\left( -\frac{1}{2}\Delta_\xi + V_{\text{eff}}(\xi) \right) f(\xi) = u f(\xi), \quad (18)$$

$$V_{\text{eff}}(\xi) = V_2(\xi) + V_3(\xi), \quad (19)$$

$$V_2(\xi) = Q\chi^2(\xi) + V_{\text{gr}}(\xi), \quad V_3(\xi) = B\chi^4(\xi), \quad (20)$$

$$V_{\text{gr}}(\xi) = -Av(\xi_B) + \frac{A}{\xi} \int_0^\xi v(s)\mathrm{d}s, \quad (21)$$

where potentials $V_2$ and $V_3$ come from two- and three-particle interactions, respectively. Substituting the found solution $\chi(\xi)$, we can see that the form of potential $V_{\text{eff}}$ also depends on a chemical potential $u$ (or parameter $\nu$).

In Figure 1, we used the values $A = 10$, $B = 20$ and $Q = 1.36$. The latter one plays the role of the "critical" value (its sense is seen in Figure 2 below, with explanations at the end of the next Section). We relate the particular forms of $V_{\text{eff}}$ given in Figure 1a with the physical situations depicted in Figure 2 below. Namely, *the green curve* in Figure 1a is chosen for a liquid-like (dense) state in Figure 2, when the three-particle interaction contributes to a hard-core part of the potential at a small $\xi$. *The red curve* was constructed in the vicinity of the critical point of the first-order phase transition, when the potential $V_{\text{eff}}$ is similar to the harmonic trap. *The black curve* corresponds to the gaseous (dilute) state in the effective (truncated) gravitational potential, when the kinetic energy term dominates ($u > 0$).

In the range $\xi \geq \xi_B$, we come to the problem of a particle in the gravitational field (see the dashed curves in Figure 1a) created by the system of $\mathcal{N}$ particles:

$$\left( -\frac{1}{2}\Delta_\xi - A\frac{\mathcal{N}}{\xi} \right) f_k(\xi) = \frac{k^2}{2} f_k(\xi), \quad f_k(\xi_B) = 0, \quad f_k'(\xi_B) = f'(\xi_B), \quad (22)$$

where:

$$\mathcal{N} = \int_0^{\xi_B} \chi^2(\xi)\, \xi^2 \mathrm{d}\xi \quad (23)$$

is the total number of particles within the ball $\xi \leq \xi_B$, which determines the total mass.

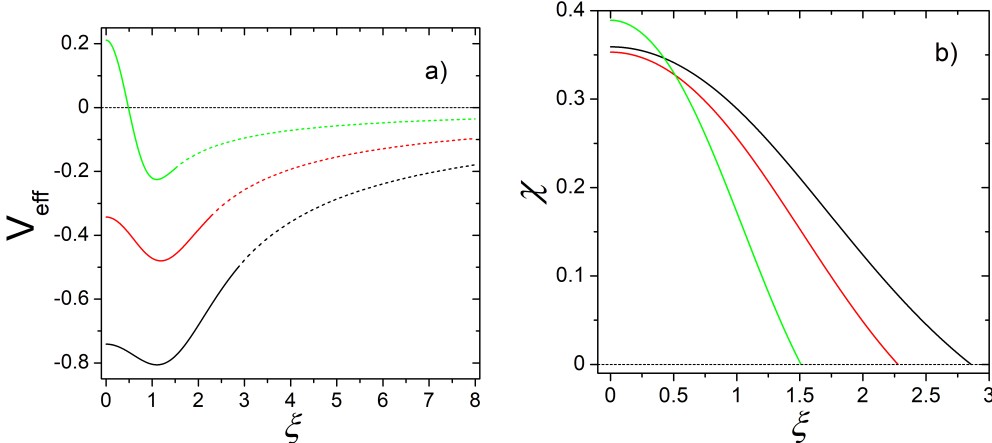

**Figure 1.** (**a**) The effective potential $V_{\text{eff}}(\xi)$ for $u_{\text{black}} \simeq 0.149$, $u_{\text{red}} \simeq -0.502$, $u_{\text{green}} \simeq -2.046$. Dashed parts of the curves correspond to a pure gravitational potential $-A \cdot \mathcal{N}/\xi$ outside the particle system; (**b**) the field $\chi(\xi)$ for the same values of chemical potential $u$, and $\chi(\xi) = 0$ at $\xi = \xi_B$. Here, $A = 10$, $B = 20$, $Q = 1.36$ for definiteness.

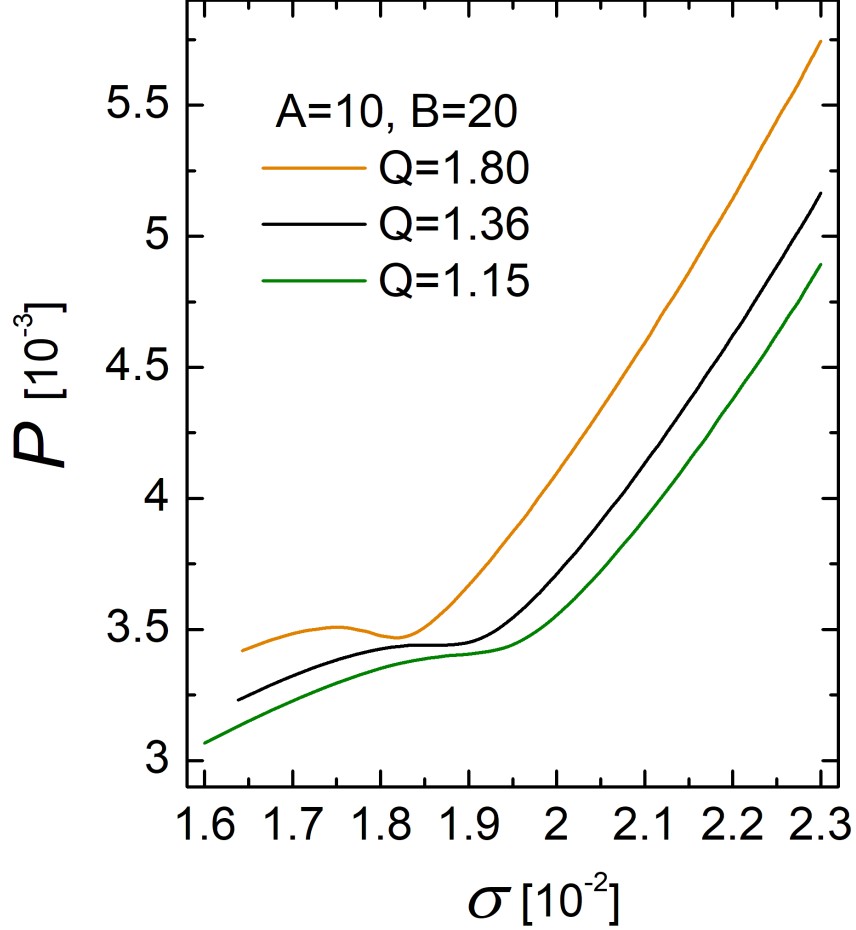

**Figure 2.** Equation of the state of dark matter at $T = 0$ and fixed parameters $A$ and $B$. A slow growth in the dimensionless pressure $P$ at a relatively low density $\sigma$ corresponds to a dilute phase of matter, while a steep rise indicates a denser liquid-like phase at high densities. An orange curve (with two extrema) demonstrates the presence of metastable states. The black line is for a critical value of $Q_c = 1.36$. The green line exemplifies a continuous transition between the two phases at $Q < Q_c$.

At this stage, the wave number $k$ in Equation (22) is ambiguous. The oscillating and decaying solution to this Equation (for any real $k$ and pure imaginary $\kappa$ and $s$) is given as

$$f_k(\xi) = \frac{c_1 M_{s,1/2}(\kappa\xi) + c_2 W_{s,1/2}(\kappa\xi)}{\xi}, \quad \kappa = 2ik, \quad s = \frac{2A\mathcal{N}}{\kappa}, \tag{24}$$

$$c_1 = f'(\xi_B)\,\xi_B^2\,\frac{W_{s,1/2}(\kappa\xi_B)}{F(\kappa)}, \qquad c_2 = -f'(\xi_B)\,\xi_B^2\,\frac{M_{s,1/2}(\kappa\xi_B)}{F(\kappa)}, \tag{25}$$

$$F(\kappa) = (1+s)M_{1+s,1/2}(\kappa\xi_B)\,W_{s,1/2}(\kappa\xi_B) + M_{s,1/2}(\kappa\xi_B)\,W_{1+s,1/2}(\kappa\xi_B). \tag{26}$$

Here, $M_{\mu,\nu}(z)$ and $W_{\mu,\nu}(z)$ are the Whittaker functions.

This solution might be interpreted as the gravitational capture of a dark matter particle (a radial wave with energy $k^2/2$) outside the dark matter ball, when $\xi \to \infty$. In the case of the initially resting particle with $k = 0$, the solution is described in terms of Bessel functions $J_1(z)$ and $Y_1(z)$.

Thus, the account of interaction confirms the possibility of bound states, which can manifest themselves in the form of different thermodynamic phases. Furthermore, all global quantities of the model, computed at fixed $A$, $Q$ and $B$, are supposed to be functions of the free parameter $\nu$. Therefore, dependence, say, of $a$ on $b$, should be treated in parametric form: $a(b) = \{(b(\nu), a(\nu))|\nu \geq \nu_{\min}\}$.

## 3. Thermodynamic Quantities and Two Phases

To study the macroscopic properties, let us define an effective chemical potential $\mu(\xi)$ [19], which includes the gravitational potential jointly with the term of quantum fluctuations, and further replaces the constant chemical potential $u$, that is:

$$\mu(\xi) + A\varphi(\xi) - \frac{1}{2\chi(\xi)}\Delta_\xi\chi(\xi) = u. \tag{27}$$

According to the equation of motion (9), $\mu$ determines $\chi$ as

$$\mu(\xi) = Q\chi^2(\xi) + B\chi^4(\xi), \qquad \mu(\xi_B) = 0. \tag{28}$$

For finding macroscopic characteristics, we consider the thermodynamic relations at $T = 0$, using a local particle density $\eta(\xi) = \chi^2(\xi)$:

$$\mathrm{d}p(\xi) = \eta(\xi)\,\mathrm{d}\mu(\xi), \qquad p(\xi_B) = 0, \tag{29}$$

$$\varepsilon(\xi) = \eta(\xi)\,\mu(\xi) - p(\xi), \qquad \varepsilon(\xi_B) = 0, \tag{30}$$

where functions $p(\xi)$ and $\varepsilon(\xi)$ determine the (dimensionless) mean pressure $P$ and the internal energy $E$:

$$P = \frac{3}{\xi_B^3}\int_0^{\xi_B} p(\xi)\,\xi^2\,\mathrm{d}\xi, \qquad E = \int_0^{\xi_B} \varepsilon(\xi)\,\xi^2\,\mathrm{d}\xi. \tag{31}$$

Hereafter, $\xi_B^3/3$ represents the volume of the system.

Therefore, we need to integrate the Gibbs–Duhem relation (29) and then substitute $p(\xi)$ into the Euler relation (30) in order to find $\varepsilon(\xi)$. This way, the explicit expressions are obtained:

$$p(\xi) = \frac{Q}{2}\eta^2(\xi) + \frac{2}{3}B\eta^3(\xi), \qquad \varepsilon(\xi) = \frac{Q}{2}\eta^2(\xi) + \frac{1}{3}B\eta^3(\xi), \tag{32}$$

which give us the equation of state by inserting the solution $\eta(\xi)$ of (12)–(14).

The (local) internal pressure $p(\xi)$ behaves spatially with a radius $\xi$ according to

$$\frac{\partial_\xi p(\xi)}{\eta(\xi)} = \partial_\xi\mu_q(\xi) = -A\frac{n(\xi)}{\xi^2} + \partial_\xi\left(\frac{1}{2\chi(\xi)}\Delta_\xi\chi(\xi)\right), \quad n(\xi) = \int_0^\xi \eta(s)\,s^2\,\mathrm{d}s. \tag{33}$$

We do not solve it because all necessary fields are found explicitly from Equations (12)–(14). Mean particle density is:

$$\sigma = \frac{3}{\xi_B^3} \int_0^{\xi_B} \chi^2(\xi)\, \xi^2 \mathrm{d}\xi. \tag{34}$$

The dependence of the mean internal pressure $P$, as can be seen in (31), on this density $\sigma$, is presented in Figure 2 for various parameters of interaction. The numerically obtained curves reveal the presence of two stable phases of dark matter with $\partial P / \partial \sigma > 0$: the dilute one ($\partial^2 P / \partial \sigma^2 < 0$) and the denser liquid-like one ($\partial^2 P / \partial \sigma^2 > 0$). The characteristic points of the phase diagram are determined from the conditions: $\partial P / \partial \sigma = 0$ and $\partial^2 P / \partial \sigma^2 = 0$. The simultaneous fulfillment of these conditions at a single point determines the critical point, which belongs to the black curve in Figure 2, and the inferring of which is one of the main tasks here. The existence of a critical point of a first-order phase transition is essentially related to the competition between gravitational and pair interactions, controlled by the parameter $Q$.

While the presence of metastable states, linked with the two-extremum behavior at $Q > 1.36$ and shown by the orange curve, clearly indicates the mixing of the gaseous and liquid phases, a detailed description of the transition between the two phases at $Q < 1.36$ requires the use of additional thermodynamic characteristics, as was already performed in [18] by means of the *perturbation pressure* $\Pi_\nu$ for $Q = 0$.

It is important to emphasize that herein we reveal a discontinuous behavior (jump between two phases) of the density $\sigma$ with the change in the *internal pressure P*. The existence of such a regime, as we show, is allowed at $Q > 1.36$ for $A = 10$ and $B = 20$. Although the existence of a denser, liquid-like phase of DM (in a relatively small region of halo) is possible at $Q = 0$, the condition $Q > 0$ is required in describing the galactic DM halos [7], even without taking into account the three-particle interaction.

Note that a continuous change in the parameter $Q$ can lead to the intersection of different curves $P(\sigma)$, which indicates the possibility of realizing one thermodynamic state by the fixation of different sets of parameters. Such ambiguity in the set of parameters can complicate the interpretation and reproduction of observables.

## 4. The Rotation Curves

Important information about dark matter is extracted and verified from the rotation curves of galaxies. As our model is aimed to study the processes in dark matter in the galaxy cores, that is, in relatively small regions of space with a significant density, its direct application to the description of rotation curves should be limited to dwarf galaxies (or other dark matter-dominated compact galaxies). One of the possibilities for describing larger (halo-type) objects is to extend the model, similarly to what was proposed in [18]. In addition, the lack of accounting for the rigid rotation of matter in the model does not allow us to describe the curves of rotating galaxies. Nevertheless, we consider it important and interesting to demonstrate the capabilities of our model, in which we accurately took into account quantum fluctuations in the condensate and (both two- and) three-particle interactions. Although the extensions of the model are indicated in [17,18], the included effects make it possible to further validate the Bose-condensate approach, to compare with and supplement the results of [7].

Thus, the tangential velocity $v$ of a test particle moving in the spherically symmetric DM halo can be represented as

$$v(r) = \sqrt{G\frac{M(r)}{r}}, \qquad M(r) = 4\pi \int_0^r \rho(s)\, s^2\, \mathrm{d}s, \tag{35}$$

where $\rho(r) = m|\psi(r)|^2$ is the mass density such that $\rho(R) = 0$ at $R = r_0 \xi_B$.

Let us illustrate the model predictions for the rotation curves of the M81 galaxy, shown in Figure 3. Note the free parameters of the model $A$, $B$, $Q$, $\nu$, $\rho_0$ that we use for fitting

along with the two restricting characteristics: the total mass $M$ of dark matter and the halo radius $R$.

The galaxy M81 with no rigid rotation gives us the most striking proof of the Bose-condensate approach, as noted in [7]. Let us use this example to emphasize the main features of the description of rotation curves.

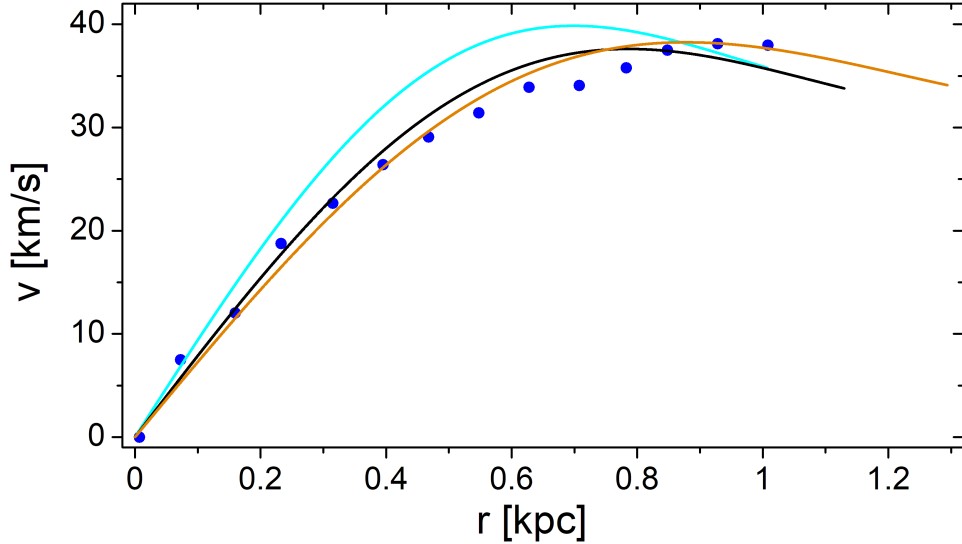

**Figure 3.** Rotation curves for the dwarf galaxy M81dwB with mass $M = 3 \times 10^8 M_\odot$. Blue dots are the observed data for $r \leq 1$ kpc. Solid lines are obtained within the model under certain restrictions: cyan line is for $M_{BEC}^{(1)} = 3 \times 10^8 M_\odot$ and $R^{(1)} = 1$ kpc; black line is for $M_{BEC}^{(2)} = 3 \times 10^8 M_\odot$ and $R^{(2)} \simeq 1.12$ kpc; orange line corresponds to $M_{BEC}^{(3)} = 3.5 \times 10^8 M_\odot$ and $R^{(3)} = 1.3$ kpc.

First of all, note that it looks rather difficult to indicate unambiguously the parameters for the rotation curve: the same dependence $v(r)$ can be realized for different sets of parameters. This is a consequence of the symmetry of the model equations. For this reason, we only give graphs that are interesting for physics, and omit the mathematics of identifying the symmetries.

In our example, it is worth noting a possibility to construct the curve colored in cyan in Figure 3 with $Q = 0$, when the pair interaction is absent. At the same time, the best fitted dependencies, represented by other curves, always require $Q > 0$. This implies that the repulsive pair interaction should be taken into account in realistic models of dark matter in (dwarf) galaxies [11,20].

## 5. Entanglement Entropy

The possibility of estimating the entanglement entropy in a general Bose condensate was developed, in particular, in [21]. The idea of using this entropy as a criterion for differentiating the BEC dark matter from cold dark matter (CDM) was proposed in [22]. Although a more precise analysis requires studying the effects of the interference of particles from interacting subsystems, here we estimate the entropy between two separated (central and surrounding) parts of radially inhomogeneous BEC dark matter of a galaxy. Omitting the normalization constants, we use the formula (in accordance with [21,22]):

$$S_E(x) = \ln \left\{ c(x\xi_B) \left[ 1 - c(x\xi_B) \right] \right\}, \qquad x = \xi/\xi_B = r/R. \qquad (36)$$

Here, $c(\xi) = n(\xi)/\mathcal{N}$ is the fraction of particles contained in the central subsystem, $n(\xi)$ being defined in (33).

A typical dependence of the entanglement entropy on the specific radius is shown in Figure 4. It is worth noting that, although the system is inhomogeneous, and $c(\xi)$ is not a constant, the maximum influence of one subsystem of particles upon the other turns out

to occur at $x = 0.5$. What concerns the negative sign of $S_E(x)$: that is eliminated by the explicit account of the normalization constant, i.e., by adding a definite positive number (of the order $10^2$ or higher, as can be seen, e.g., in [22]), which depends on the mass of dark matter particles and their number in the subregion.

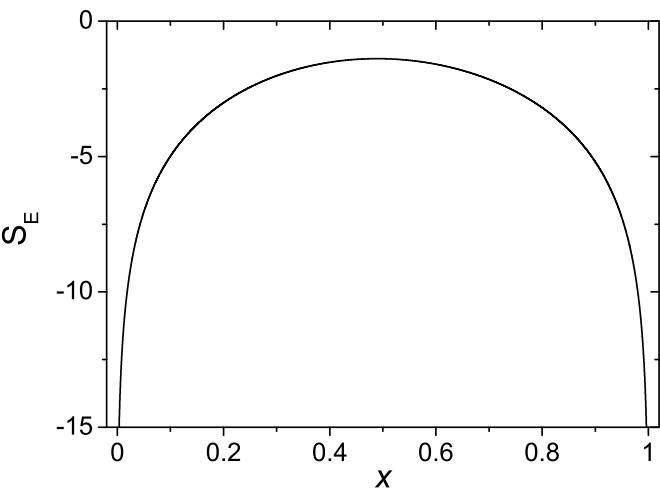

**Figure 4.** Entanglement entropy $S_E$ within the model with $A = 10$, $B = 20$, $Q = 1.36$ and $u \simeq -2.046$.

## 6. Discussion

In the framework of the extended version of the BEC dark matter model that involves two- and three-particle interactions, we explored the solutions of the system of Equations (11)–(14) as well as the properties of thermodynamic functions. The analysis has led us to the conclusions that the interplay of the basic free parameters $A$, $Q$ and $B$ gives interesting consequences for the system under study including dark matter halo properties: (i) first, the effective interaction potential, see (18)–(21) and Figure 1, and the density profiles; (ii) the equation of state which shows basic dependence on the parameter $Q$ including the existence of its "critical" value $Q_C$ that separates different regimes; (iii) persistence at $Q > Q_C$ of the nontrivial phase structure (inherited from the restricted pure $\psi^6$ sub-model). The obtained information on the thermodynamic function such as the mass density profile, enabled us to infer in Section 3 the galactic rotation curves, and the bipartite entanglement entropy in Section 4. The former clearly demonstrated, with the example of dwarf galaxy M81, that the model can easily provide nice agreement with observational data. As follows from the treatment of rotational curves, the parameter $Q$ responsible for the two-particle interaction plays an essential role. In addition, as seen in Figure 2, at $Q > Q_C$, the model reveals the rather rich behavior: the curves possess two extrema that imply the metastability region. At $Q \geq Q_C$, only inflection points survive. Without $Q$ (without pair interaction), we could not have such features of unexpected behavior. Anyway, for both $Q > 0$ and $Q = 0$ (that brings us back to the situation explored in [18]), the two phases are present, though their identification involves differing thermodynamical functions.

It is of interest to compare the model considered in this paper with some of the nonlinear (namely, non-polynomial) extensions of the BEC model of DM, e.g., those studied in [23–26] where the scalar potential $V_0[\cosh(\lambda\kappa\Phi) - 1]$ was employed, with $\kappa = \sqrt{8\pi G}$. The essential feature of these models is the presence of all-order nonlinearities, with the corresponding powers of the single free parameter $\lambda$ (note that $\lambda$ can be viewed as a deformation parameter since at $\lambda \to 0$, the potential and thus the interaction vanish). On one hand, the model studied above is obviously simpler than the models involving cosh or cos: indeed, we encountered the first terms of the series expansion if the parameters are restricted as $A = \sqrt{B} = \lambda$. On the other hand, the $\sim \psi^4$ plus $\sim \psi^6$ model studied herein is richer in the sense that it operates with the free parameters $A$, $B$, $Q$, instead of a single one. It is this property that allowed us to disclose (confirm) the existence of two phases and of

the phase transition. Moreover, the influence of different values of these parameters was essential in our treatment of the rotation curves and the entanglement entropy, as can be seen in Figure 4 above.

Concerning the entanglement, an interesting question arises for the situation when dark matter particles are not elementary bosons but composites built from two bosons or two fermions. In both cases: (i) the composites differ from pure bosons and are naturally realizable, as can be seen in [27], through deformed oscillators or deformed bosons (or quasibosons); (ii) bipartite internal entanglement entropy of quasibosons obtained in [28,29] turned out to depend on the deformation parameter. The question now is as follows: to what extent does the microscopic intra-quasibosonic entanglement and its entropy affect (superimpose with) the macroscopic entanglement and the entanglement entropy that was studied in Section 5 above. Furthermore, it is no doubt important to explore in detail the connection [21,30–32] between the peculiarities of the behavior of entanglement and the phase transition (of the first order in our case), especially in the context of the properties of dark matter. We hope to explore these questions in one of our future works.

**Author Contributions:** The contributions of both authors to the article preparation were equal. Conceptualization and draft preparation, A.M.G. and A.V.N.; numerical analysis, A.V.N. All authors have read and agreed to the published version of the manuscript.

**Funding:** This research received no external funding.

**Acknowledgments:** A.M.G. acknowledges support from the National Academy of Sciences of Ukraine by its priority project No. 0120U100935 "Fundamental properties of the matter in the relativistic collisions of nuclei and in the early Universe". The work of A.V.N. was supported by the project No. 0117U000238 of NAS of Ukraine.

**Conflicts of Interest:** The authors declare no conflict of interest.

## Note

1     See Figure 1b for the behavior of $\chi(\xi)$.

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
