# Peer review of "Phases of the Bose–Einstein Condensate Dark Matter Model with Both Two- and Three-Particle Interactions"

_universe, doi:10.3390/universe7100359_

Round 1

Reviewer 1 Report

In this paper the authors consider a 6th order Bose-Einstein condensate (BEC) dark matter model as a continuation of their previous work published in PRD. In the current paper, they include the 4th order attractive self-interaction and present the results in the context of galaxy rotation curves and entanglement entropy.

I find the article to be suitable for publication in Universe but I feel it is important that some parts of it are rewritten (or phrased in another words) so that they don't repeat word for word the PRD work. For example: 65-74 - almost 1 to 1 repetition to PRD, lines 77-79 too, 86-87 too, 94-96 (total repetition with 1 word changed modes -> options).

Also I have some questions and comments:

1) in Eq. 11, is the terms ~Q_*/3 or Q_*/2 (in (6) it's Q/2). In line 79, it might be worth mentioning how the field equations are obtained for people who haven't read the PRD paper.

2) Line 102 "primary information" sounds odd to me, maybe "important or critical information".

3) Line 109, it's not obvious how the critical value is chosen so might be good to explain. You point to the figure, but as I assume Q is continuous it's not obvious why it's exactly this number (or I may have missed it in the text).

4) Line 133 or 134, probably the citation for writing the chemical potential in this form ([52] from the PRD paper) should be added. As a whole, the article has very few citations so I suggest adding some more - the PRD paper has 74 citations, this one has only 27 . This suggestion is optional but I believe it will improve the quality of the article. 

5) Line 238 "From one hand" -> "On one hand". 

6) Also, it might be good to explain somewhere (the conclusion?) what is the final difference with and without the \Phi^4 term as the only comparable plots I saw were Fig. 2 vs. Fig. 6 in the PRD paper and they look very similarly.  You mentioned that now there are 3 parameters, instead of 2,  but that's not always an advantage and also you say that Q "plays definite role, but not the decisive one." So maybe elaborating with one sentence will be nice.

Reviewer 2 Report

The manuscript is written at a high professional level. It deserves to be published in the journal "Universe". Two minor comments will only improve the manuscript, I think.

Reviewer 3 Report

See Report.

Author Response

There were no remarks/suggestions in this Reviewer's report